# Optimization and Validation of a Classification Algorithm for Assessment of Physical Activity in Hospitalized Patients

**DOI:** 10.3390/s21051652

**Published:** 2021-02-27

**Authors:** Hanneke C. van Dijk-Huisman, Wouter Bijnens, Rachel Senden, Johannes M. N. Essers, Kenneth Meijer, Jos Aarts, Antoine F. Lenssen

**Affiliations:** 1Department of Physical Therapy, Maastricht University Medical Center, 6229 HX Maastricht, The Netherlands; rachel.senden@mumc.nl (R.S.); af.lenssen@mumc.nl (A.F.L.); 2CAPHRI School for Public Health and Primary Care, Maastricht University, 6200 MD Maastricht, The Netherlands; 3Instrument Development, Engineering and Evaluation, Maastricht University, 6200 MD Maastricht, The Netherlands; wouter.bijnens@maastrichtuniversity.nl (W.B.); jos.aarts@maastrichtuniversity.nl (J.A.); 4Department of Nutrition and Movement Sciences, NUTRIM, Maastricht University, 6200 MD Maastricht, The Netherlands; hans.essers@maastrichtuniversity.nl (J.M.N.E.); kenneth.meijer@maastrichtuniversity.nl (K.M.)

**Keywords:** physical activity, accelerometers, algorithm, validation, hospitalized patients

## Abstract

Low amounts of physical activity (PA) and prolonged periods of sedentary activity are common in hospitalized patients. Objective PA monitoring is needed to prevent the negative effects of inactivity, but a suitable algorithm is lacking. The aim of this study is to optimize and validate a classification algorithm that discriminates between sedentary, standing, and dynamic activities, and records postural transitions in hospitalized patients under free-living conditions. Optimization and validation in comparison to video analysis were performed in orthopedic and acutely hospitalized elderly patients with an accelerometer worn on the upper leg. Data segmentation window size (WS), amount of PA threshold (PA Th) and sensor orientation threshold (SO Th) were optimized in 25 patients, validation was performed in another 25. Sensitivity, specificity, accuracy, and (absolute) percentage error were used to assess the algorithm’s performance. Optimization resulted in the best performance with parameter settings: WS 4 s, PA Th 4.3 counts per second, SO Th 0.8 g. Validation showed that all activities were classified within acceptable limits (>80% sensitivity, specificity and accuracy, ±10% error), except for the classification of standing activity. As patients need to increase their PA and interrupt sedentary behavior, the algorithm is suitable for classifying PA in hospitalized patients.

## 1. Introduction

Low amounts of physical activity (PA) and prolonged periods of uninterrupted sedentary activity are common in hospitalized patients. Patients spend between 87% and 100% of their day lying in bed or sitting in a chair [1,2,3,4,5]. Little time is spent being active, and bouts of standing and walking are usually short [6,7]. This sedentary behavior is found in all patient subpopulations. On average, surgical inpatients spend 10 to 71 min per day standing and walking [1,8,9,10], compared to 66 to 117 min for geriatric inpatients [7,11,12,13], 1 to 184 min for medical inpatients [14,15,16,17,18,19,20], 10 to 86 min for post-stroke inpatients [21,22,23,24,25,26], and 0 min for patients admitted to the intensive care unit [27,28].

Low amounts of PA and prolonged periods of uninterrupted sedentary activity during hospitalization have been associated with functional decline [29,30], a decline in physical performance [30], increased insulin resistance [30], increased length of stay [31]_,_ increased risk of institutionalization [16], and mortality [29,32,33,34]. To reduce the risks of these negative effects, interventions aimed at increasing the amounts of PA and breaking up prolonged periods of sedentary activity are essential [30,35,36,37,38,39]. In order to support (i.e., perform and/or evaluate) such interventions, it is necessary to measure patients’ PA behavior in an objective and accurate way [2,40,41].

Monitoring patients’ PA behavior during hospitalization is commonly performed using self-reported measures, behavior mapping, or wearable activity monitors [1,2,40,42]. Self-reported measures (e.g., surveys or diaries) are subjective and show low validity and reliability [40,42,43]. Behavior mapping involves direct, structured observation and classification of patients’ PA behavior by observers [44,45]. This is labor-intensive and may intrude upon patients’ privacy [41,46]. Moreover, it may under- or overestimate amounts of PA and periods of uninterrupted sedentary activity when observations are performed during daytime hours only, or when sampled at brief intervals (e.g., one minute in every ten minutes) [16,41,46,47,48]. As bouts of walking often last less than two minutes, they might not be recorded, resulting in an underestimation of the amount of PA. Wearable activity monitors, such as accelerometers, allow for objective, continuous quantification and classification of patients’ PA behavior over longer time periods, with minimal effort and invasiveness [1,2,6,41,49,50]. Despite all their advantages, accelerometers have not been widely integrated in clinical practice, due to issues relating to feasibility, reliability, and validity [2,41,50]. Accelerometers measure raw accelerations obtained from movements of a body or a body segment. PA behavior is then estimated by applying an algorithm to the raw data [51]. Most algorithms are built with the same conceptual building blocks, viz., (1) a pre-processing phase to remove artifacts from the raw data, (2) data segmentation, (3) extraction of data features, and (4) a classifier that translates the raw data into interpretable outcome measures [52,53,54,55,56,57].

The performance metrics of an algorithm to measure patients’ PA behavior are influenced by patient characteristics (e.g., age, walking speed, gait pattern, and the use of a walking aid), sensor wear location, number of sensors used, and outcome parameters (e.g., classifying activities, step count, and intensity) [41,42]. Time spent in dynamic activities (e.g., walking, stair climbing) and the classification of postural transitions from sedentary to upright position are the most relevant outcome parameters for hospitalized patients, as they need to increase their amount of PA and interrupt prolonged periods of sedentary activity [1,35,45]. Most accelerometer algorithms are validated in healthy adults and lack the sensitivity to classify slow or impaired gait [58,59]. They are not able to accurately differentiate slow gait and shuffling from standing. However, slow and impaired gait, as well as the frequent use of walking aids, are common in hospitalized patients. As a result, using an algorithm that is validated in healthy adults in a population of hospitalized patients would require optimization and validation of the algorithm’s performance [41]. Previous studies have shown that the validity of existing algorithms to discriminate between sedentary, standing, and dynamic activities, and to classify postural transitions in hospitalized patients, varies and is usually investigated in small study samples [12,35,45,46,48,55,60,61,62]. A suitable algorithm for hospitalized patients that is able to discriminate between standing and dynamic activities, as well as to classify postural transitions, is currently lacking [63].

Recently, Hospital Fit (HFITAPP0, Maastricht Instruments B.V., Maastricht, The Netherlands), a smartphone application connected to an accelerometer, was developed to enable PA monitoring and to stimulate recovery in hospitalized patients [1]. The algorithm embedded in this accelerometer is able to differentiate time spent being sedentary (lying/sitting) from time spent being active (standing/dynamic) in hospitalized patients. The current study is built upon Hospital Fit by aiming to discriminate between standing and dynamic activities and by classifying postural transitions. Bijnens et al. have presented an adjustable PA classification algorithm that is validated to discriminate between sedentary, standing, and dynamic activities in healthy elderly persons [49]. Its easily adjustable parameters enable the performance of this algorithm to be optimized for different target populations and sensor wear locations. The algorithm had not yet been optimized or validated in hospitalized patients. Doing so and implementing the proposed algorithm in Hospital Fit would improve PA monitoring in hospitalized patients. The aim of this study was therefore to optimize and validate a PA classification algorithm which is able to discriminate between sedentary, standing, and dynamic activities, and to detect postural transitions among hospitalized patients. We assessed the concurrent validity of the algorithm to classify sedentary, standing, and dynamic activities and detect postural transitions in hospitalized patients, by checking it against video analysis.

## 2. Materials and Methods

### 2.1. Study Design

This single-center, prospective validation study was conducted at Maastricht University Medical Center (MUMC+) in Maastricht, The Netherlands, between November 2019 and March 2020.

### 2.2. Study Population

Patients who received physical therapy and were (1) admitted for elective total knee arthroplasty (TKA) or total hip arthroplasty (THA) at the Department of Orthopedic Surgery and Traumatology, or (2) aged 70 years or older and acutely hospitalized at the Department of Internal and Geriatric Medicine at the MUMC+ were invited to participate. Patients were recruited during weekdays. Patients scheduled for elective TKA or THA received verbal and written information about the study from their physical therapist four to six weeks prior to surgery, during preoperative screening. A researcher contacted the patients during their hospitalization, and written informed consent was obtained before they entered the study. Acutely hospitalized elderly patients received verbal and written information about the study from their physical therapist during their first physical therapy session. Informing these patients prior to hospitalization was not possible because they were admitted acutely. A researcher contacted the patients the next day. If patients were interested in participating, an informed consent form was provided by the researcher and written informed consent was obtained before they entered the study. Informed consent was signed in the patient’s own room. Confidential processing of data and anonymity were guaranteed.

Patients were eligible if they met the following inclusion criteria: receiving physical therapy, aged 18 years or older and admitted for TKA or THA at the Department of Orthopedic Surgery and Traumatology, or aged 70 years or older and acutely admitted at the Department of Internal and Geriatric Medicine, having been able to walk independently two weeks prior to admission as scored on the Functional Ambulation Categories (FAC > 3) [64], and having a sufficient understanding of the Dutch language. Exclusion criteria were: the presence of contraindications to walking or wearing an accelerometer on the upper leg, admission to the intensive care unit, impaired cognition (delirium / dementia) or being incapacitated as reported by the attending doctor, a life expectancy of less than three months, and previous participation in this study.

This study was performed in compliance with the Declaration of Helsinki and was approved by the Medical Ethics Committee of the University Hospital Maastricht and Maastricht University (METC azM/UM), registration number 2019-1265.

### 2.3. Data Collection

Fifty patients were enrolled after signing the informed consent. The sample size of 50 corresponds to that used in previous validation studies, which included 8 to 99 participants [35,42,46,62,65,66,67,68,69,70]. Optimization of the adjustable algorithm described by Bijnens et al. was performed on data of 25 patients [49]. Validation of the optimized algorithm was then performed on data of the remaining 25 patients. After inclusion, patients were randomized 1:1 to the optimization or validation group, using a stratified block randomization. To ensure an equal distribution of orthopedic and elderly patients within the optimization and validation groups, patients were first stratified by department (“Orthopedic Surgery and Traumatology’ or ‘Internal and Geriatric Medicine”) before they were randomized (Figure 1). The randomization and allocation of patients was carried out by an independent researcher. The randomization schedule was created using a computer-based random number generator. Medical and demographic data (age, sex, and use of a walking aid) were extracted from the electronic patient records. Missing data were not substituted and drop-outs were not replaced.

All patients received a referral to usual care physical therapy from their physician. As physical therapy sessions often comprise a significant part of the patients’ PA behavior during their hospital stay, a randomly selected physical therapy session was used to collect data under free-living conditions. This could range from the first to the last physical therapy session, which enabled the performance of the algorithm to be investigated in a variety of patients with different gait patterns. Physical therapy sessions were aimed at increasing PA and stimulating functional recovery of activities of daily living which are essential in order to function independently at home. Sedentary, standing, and dynamic activities (e.g., walking, stair climbing) as well as postural transitions from sedentary (sitting/lying) to upright (standing/dynamic) positions were performed at least once during each physical therapy session. The exact content of physical therapy sessions depended on the diagnosis and needs of the individual patients. The order, pace, and duration of activities varied between individuals. If necessary, patients used a walking aid. This study did not interfere with the content of the physical therapy sessions.

#### 2.3.1. Video Recordings

Patients were recorded from the waist down using a handheld camera (HDC-HS60, Panasonic, Osaka, Japan). Recording the faces or other people within the hospital wards was avoided. The video recordings served as a reference for the classification of sedentary, standing, and dynamic activities, as well as for the detection of postural transitions. Video recording was used as the gold standard in activity monitoring, as it allows the most accurate activity classification, and offers the possibility to reanalyze data by single or multiple observers [71,72,73]. After the physical therapy session, the video recordings were uploaded to a computer.

#### 2.3.2. Acceleration Data

Acceleration data were acquired with a MOX Activity Logger (MOX; Maastricht Instruments, Maastricht, The Netherlands (Figure 2A)). The MOX contains a tri-axial accelerometer sensor (ADXL362; Analog Devices, Norwood, MA, USA). The small, lightweight, waterproof device (35 × 35 × 10 mm, 11 g) measures raw acceleration data (±8 g) for three orthogonal sensor axes (X, Y, and Z) at a 25 HZ sampling rate, and stores the data directly in its internal memory. Each axis is factory-calibrated against gravity. The MOX is capable of measuring and storing data continuously for up to seven days. Data analysis is performed offline. After uploading the raw acceleration data provided by the MOX to a computer, an algorithm can be applied to these raw data. The MOX has been successfully used as an activity logger for PA monitoring in colorectal cancer survivors, chronic organ failure patients, total knee and hip arthroplasty patients, and healthy elderly subjects [1,40,49,74,75].

The MOX uses a custom-made, double-sided, waterproof, hypoallergenic patch for body attachment. Prior to the physical therapy session, this patch was used to attach the MOX to the upper leg (ten centimeters proximal of the patella, Figure 2B). The upper leg location was chosen as it allows for classification of body postures and movements (e.g., lying/sitting, standing, walking) [76,77,78]. For the orthopedic patients, the MOX was attached to the non-operated leg. For the acutely hospitalized elderly patients, the MOX was attached to the right leg. Both at the beginning and the end of the physical therapy session, the researcher tapped the MOX twice for the purpose of post-hoc synchronization between the video recording and the raw acceleration data. After the treatment session, the MOX was removed and the raw acceleration data were uploaded to a computer via a USB connection.

### 2.4. Data Analysis

#### 2.4.1. Video Recordings

All video recordings were continuously classified as (1) sedentary, (2) standing, or (3) dynamic activities using the Behavioral Observation Research Software (BORIS, v7.9.19) [79]. Postural transitions were recorded when a sedentary activity was followed by a standing or dynamic activity. Three trained observers (R.S., H.C.v.D.-H., J.M.N.E.) were given clear definitions to classify each activity or transition (Table 1).

Each video recording was independently analyzed by two observers. In order to minimize bias, different combinations of observers were used. Observers were blinded to the classifications made by other observers and by the algorithm. Using video recordings as a gold standard requires high inter-observer reliability. This was assessed based on the total time per activity per patient, using the intraclass correlation coefficient (ICC, two-way random, absolute agreement). An ICC ≥ 0.9 was considered high [80].

#### 2.4.2. Algorithm Optimization

The adjustable classification algorithm previously described by Bijnens et al. [49] was used as the starting point for the optimization process. This algorithm contains three parameters that can be easily adjusted for target population and sensor wear location: (1) data segmentation window size (WS), (2) amount of physical activity threshold (PA Th), and (3) sensor orientation threshold (SO Th). The algorithm was recently validated to discriminate between sedentary, standing, and dynamic activities in healthy elderly persons with an upper leg wear location. The parameter settings of this algorithm were referred to as MOXAL (WS: 2 s, PA Th: 7 counts per second (cps), SO Th: 0.8 g) [49].

To determine the performance of MOXAL in hospitalized patients with an upper leg wear location, we applied it to the raw acceleration data of our optimization group. MATLAB (R2018b; The MathWorks Inc., Natick, MA, USA) was used to convert the raw acceleration data into classifications of sedentary, standing, or dynamic activities for each data segmentation window.

The classification accuracy of the algorithm was assessed by calculating sensitivity, specificity, and accuracy for each activity [81]. The acceleration data were manually synchronized with the data of the video recordings. Data of the video recordings were segmented into windows of similar length as the algorithm’s data segmentation window size, in order for it to be used as a reference. The main activity within each window was used as a comparator. For each individual, activity classifications derived from MOXAL were compared with classifications derived from the video recordings in a confusion matrix. Comparisons were made for each window within the entire measurement period. The confusion matrix showed how often activity classifications were detected correctly by the algorithm in comparison with the video classifications, and how often activities were classified differently. Confusion matrices were derived for sedentary, standing, and dynamic activities as described by Ruuska et al. [81]. Figure 3 provides an example of a confusion matrix for dynamic activity. To assess the performance accuracy for postural transitions, a synchronized time array was created for the annotated video data and algorithm classifications, in order to create a confusion matrix. In this time array, a sedentary window followed by a standing or dynamic window was given the value “one,” whereas adjacent windows of the same activity were given the value “zero.” Sensitivity, specificity, and accuracy were subsequently calculated per activity and for postural transitions (Equations (A1)–(A3)) in Appendix B [81]. Additionally, the classification accuracy was calculated over all activities (total), based on the sum of the confusion matrices of the separate activities.

To assess the classification error of the algorithm, percentage error (PE) and absolute percentage error (APE) were calculated per activity (Equations (A4) and (A5)) [49]. PE and APE reflect the error between the video recordings and the algorithm, and were assessed based on the total time per activity as classified by the video recordings. To assess the error of postural transitions, the total numbers of postural transitions determined by the video classifications and the algorithm were compared. A negative PE value reflects an overestimation by the algorithm, while a positive PE value reflects an underestimation. APE does not differentiate between over- or underestimation, and thus provides an indication of the magnitude of the error. As PE and APE are relative measures, it is possible to compare them across studies [51]. Additionally, the errors over all activities (total) were calculated as the sum of the errors of the separate activities.

All performance metrics of the classification accuracy and error were determined for each individual, and medians (Q1 to Q3) were calculated per group. The median and interquartile ranges were used to present non-normally distributed data. Sensitivity, specificity, and accuracy values of 80% or higher were considered acceptable [71,82]. PE ± 10% and APE lower than 10% were considered to be within acceptable limits [83,84].

During the optimization phase, the parameter settings of MOXAL (WS, PA Th, and SO Th) were adjusted to reduce the total activity APE. Out of a set of 4025 combinations (WS ranging from 0.4 s to 10 s in steps of 0.4 s, PA Th ranging from 2 cps to 6 cps in steps of 0.025 cps), the parameter settings resulting in the lowest total activity APE were referred to as MOXAL_Opt_ (WS: 0.8 s, PA Th: 3.85 cps, SO Th: 0.8 g). The performance metrics (sensitivity, specificity, accuracy, PE, and APE) of MOXAL_Opt_ were assessed in the same way as for MOXAL. As the optimization did not sufficiently improve the performance of the algorithm, additional modifications had to be introduced.

Since the amount of PA for dynamic activity was very low for the hospitalized patients, there was a relatively small difference in the amount of PA between standing and dynamic activities. This small difference made it challenging to find an appropriate PA Th. Therefore, additional modifications were introduced regarding the decision tree and the calculation of the amount of PA. The decision tree was modified to first discriminate between sedentary and upright windows based on the SO Th. Next, the upright windows were further classified as standing or dynamic activity based on the PA Th. Furthermore, in MOXAL and MOXAL_Opt_, the amount of PA was calculated by combining the raw acceleration data of the three orthogonal sensor axes. In the modified algorithm, only the most sensitive axis was used, to avoid masking effects of other axes and improve the calculation of the amount of PA. Walking produces a distinct pattern in both anterior-posterior and vertical directions. In patients who walk slowly, especially those using walking aids, the anterior-posterior acceleration signal is more pronounced than the vertical acceleration signal [55]. Using the anterior-posterior axis was therefore expected to improve the calculation of the amount of PA in hospitalized patients and consequently improve the classification of standing and dynamic activities.

After these modifications, the algorithm was optimized again by adjusting the parameter settings. Using the same 4025 combinations as before, the parameter settings resulting in the lowest total APE were referred to as HFITAL (WS: 4 s, PA Th: 4.3 cps, SO Th: 0.8 g). Next, the performance metrics of HFITAL were assessed in the same way as for MOXAL. A schematic overview of the data processing of HFITAL is shown in Figure A1.

#### 2.4.3. Algorithm Validation

After the algorithm had been optimized, it was validated by assessing the performance of the optimized algorithm in a different group of patients within the same target population. Data of the validation group were used to assess the performance metrics of HFITAL as regards classifying sedentary, standing, and dynamic activities and detecting postural transitions in hospitalized patients in comparison to the video analysis. The performance metrics (sensitivity, specificity, and accuracy, PE, and APE) were calculated in the same way as described above for the algorithm optimization. In addition, a subgroup analysis was performed in which the performance metrics were assessed for acutely hospitalized elderly patients and orthopedic patients separately, providing more insight into the performance of the algorithm in the two groups.

## 3. Results

### 3.1. Participant Characteristics

Of the 50 participating patients, four (8.0%) were excluded due to problems with synchronization or technical complications. This resulted in 46 (92.0%) patients for analysis, with 22 (47.8%) in the optimization group and 24 (52.2%) in the validation group. The baseline characteristics of patients included in the optimization and validation groups are reported in Table 2.

### 3.2. Inter-Observer Reliability

The inter-observer reliability of the classification of activities based on the video recordings was high. The ICC values for the optimization group were 1.000, 0.994, and 0.995 for sedentary, standing, and dynamic activities, respectively. The ICC values for the validation group were 1.000 for sedentary and dynamic activities, and 0.997 for standing activity.

### 3.3. Algorithm Optimization

The median (Q1 to Q3) duration of the measurement protocol for patients in the optimization group was 12.3 (8.3 to 15.0) minutes per patient. The median (Q1 to Q3) times spent performing sedentary and standing activities were 3.0 (0.7 to 7.4) and 2.1 (1.5 to 3.9) minutes per patient, respectively. The majority of time was spent performing dynamic activity, with a median (Q1 to Q3) time of 4.9 (3.9 to 6.5) minutes per patient.

Applying MOXAL to the acceleration data of the optimization group resulted in the performance metrics shown in Table 3 and Figure 4. All performance metrics are expressed as median percentages (Q1 to Q3). MOXAL resulted in a low sensitivity of 79.0% (40.1% to 92.9%) and a high APE of 18.2% (3.4% to 55.4%) for the classification of dynamic activity, as well as a high PE of −33.1% (−114.8% to 1.1%) and an APE of 34.0% (6.1% to 114.8%) for the classification of standing activity. Total APE was 18.9% (4.2% to 51.0%).

Applying MOXAL_Opt_ to the data of the optimization group resulted in a low sensitivity of 74.5% (42.3% to 88.0%) for the classification of dynamic activity and high PE values of 10.4% (6.1% to 17.4%), −42.9% (−106.8% to 1.4%), and −200.0% (−290% to −150%) for the classification of sedentary activities, standing activities, and postural transitions, respectively. None of the APE values fell within the acceptable limits. Total APE was 11.8% (8.7% to 56.0%). Since the performance metrics of MOXAL_Opt_ did not improve compared to MOXAL (in some cases they even deteriorated), additional modifications were introduced to the algorithm, resulting in the optimized algorithm HFITAL.

Applying HFITAL to the acceleration data of the optimization group resulted in acceptable performance metrics, for both the classification of sedentary, dynamic, and total activities, and for the detection of postural transitions. Only the sensitivity of 67.3% (57.1% to 76.4%), the PE of 20.2% (−10.1% to 30.5%), and the APE of 25.1% (11.8% to 35.5%) for the classification of standing activity did not fall within the acceptable limits. Total APE was 7.6% (4.8% to 15.3%).

A detailed overview of the parameter settings of the activity classification algorithms evaluated during the optimization process, and a schematic overview of the data processing of HFITAL, can be found in Table A1 and Figure A1. A graphical representation of the raw acceleration data, the video annotations, and the classification by MOXAL, MOXAL_Opt_, HFITAL is given as an example in Figure A2. Detailed numeric results can be found in Appendix A.

### 3.4. Algorithm Validation

The median (Q1 to Q3) duration of the measurement protocol for patients included in the validation group was 10.8 (7.4 to 18.4) minutes per patient. The median (Q1 to Q3) times spent performing sedentary and standing activities were 3.7 (1.8 to 6.3) and 1.9 (0.4 to 4.6) minutes per patient, respectively. The majority of time was spent performing dynamic activity, with a median (Q1 to Q3) time of 4.4 (3.8 to 7.5) minutes per patient.

Validation of the optimized algorithm was performed by applying HFITAL to the acceleration data of the validation group. This resulted in the performance metrics shown in Table 4 and Figure 5. The classification of activities and the detection of postural transitions produced sensitivity, specificity, and accuracy values above 89.2%, while APE and PE values were below 8.6%. Postural transitions were accurately detected by the algorithm, showing an identical number of transitions for 76% of the patients. In one patient, HFITAL overestimated the number of transitions by two. In four patients, HFITAL overestimated the number of transitions by one. With a sensitivity of 65.0% (34.1% to 76.9%), a PE of 21.3% (−3.9% to 50.2%) and an APE of 29.2% (14.6% to 55.2%), the classification of standing activity did not meet the acceptable limits.

Subgroup analysis of the data of the acutely hospitalized elderly patients resulted in sensitivity, specificity, and accuracy values above 88.6%, and APE and PE values below 8.2% for sedentary, dynamic, and total activities, as well as postural transitions. However, with a sensitivity of 34.7% (20.3% to 55.3%), a PE of 49.0% (13.6% to 58.6%), and an APE of 51.6% (29.2% to 61.2%), the classification of standing activity resulted in unacceptable performance metrics.

Similarly, subgroup analysis of the data of orthopedic patients resulted in sensitivity, specificity, and accuracy values above 88.69%, with APE and PE values below 9.1% for the classification of sedentary, dynamic, and total activities, as well as postural transitions. The classification of standing showed a sensitivity of 71.8% (65.7% to 81.8%) and a PE of 9.1% (−18.5% to 23.7%). However, the APE values of the classification of standing and dynamic activities were too high (18.2% [12.8% to 30.4%] and 12.9% [4.9% to 22.1%], respectively) (Table 4, Figure 5). Detailed numeric results can be found in Appendix A.

## 4. Discussion

The primary aim of this study was to present and validate an optimized PA classification algorithm (HFITAL) which is able to discriminate between sedentary, standing, and dynamic activities, and able to detect postural transitions among hospitalized patients in a free-living setting. The results show that with an accelerometer worn on the upper leg, the best classification performance for HFITAL was achieved with the following parameter settings: a data segmentation window size (WS) of 4 s, an amount of physical activity threshold (PA Th), of 4.3 cps, and a sensor orientation threshold (SO Th) of 0.8 g. Validation of HFITAL showed that the classification of sedentary and dynamic activities, as well as the detection of postural transitions, produced sensitivity, specificity, and accuracy values above 89.0% and percentage error and absolute percentage error below 8.0%. Furthermore, the performance metrics of the classification of sedentary and dynamic activities, as well as the detection of postural transitions, fell within the acceptable limits for at least 75.0% of the patients, indicating the robustness of HFITAL. With a sensitivity of 65.0%, a PE of 21.3%, and an APE of 29.2%, only the classification of standing activities did not fall within acceptable limits.

The finding that it was difficult for HFITAL to correctly classify standing activity in hospitalized patients may have resulted from patients’ slow or shuffling gait and the frequent use of walking aids. Standing as well as slow or shuffling gait are all characterized by small acceleration amplitudes. These comparable acceleration amplitudes lead to minimal differences between the amount of PA calculated for standing and dynamic activities, making it more difficult to select an appropriate PA Th to distinguish between these activities. The algorithm could thus have mistakenly classified standing activity as dynamic activity, resulting in a possible underestimation of the time classified as standing activity and an overestimation of the time classified as dynamic activity. The relatively low performance metrics for the classification of standing activity may also be explained by the relatively small amount of time spent in standing activity during the measurements, compared to the time spent in sedentary or dynamic activities. As sensitivity, specificity, and accuracy are influenced by the total measurement time per activity, a few misclassifications of standing activity could have resulted in a relatively larger effect on the performance metrics of standing compared to dynamic activity. Lastly, in order to assess the true performance of the algorithm, we refrained from excluding outliers from the analysis. All these factors may have contributed to the low median sensitivity, PE, and APE as well as the wide Q1 to Q3 for the classification of standing activity by HFITAL.

The subgroup analysis showed lower performance metrics for the classification of standing activity by HFITAL in acutely hospitalized elderly patients compared to orthopedic patients. Slow gait and the use of walking aids are common in both populations [85,86,87], which was confirmed by our video recordings. However, our recordings also showed a higher prevalence of shuffling gait in the acutely hospitalized elderly patients, including more time spent in double support, reduced step length, and reduced lifting of the feet during the swing phase of walking. These characteristics may have resulted in lower acceleration amplitudes for walking in this population, making it more difficult to correctly classify standing activity. Investigating the degree to which shuffling or slow gait contributed to the limited performance in the classification of standing activity requires further research, using a standardized protocol and including walking speed as an outcome measure.

For the optimization of the classification algorithm, we chose total activity APE as the performance metric used to select the best combination of adjustable parameters. This metric was selected to ensure that all three activity types would be correctly classified. Selecting a different performance metric could result in a different combination of parameter settings. This may improve the performance of the classification of standing activity but may possibly also negatively influence the performance of the classification of sedentary and dynamic activities. To the best of our knowledge, there is no consensus on which performance metrics should ideally be used. Further research is recommended to investigate which performance metrics are most suitable for the optimization of an adjustable PA classification algorithm.

The classification of sedentary, standing, and dynamic activities and postural transitions in hospitalized patients may be further improved by the use of a different type of classifier. Such a different type of classifier may also enable the classification of a broader range of activity types. Recently, pattern recognition and machine learning algorithms have received a great deal of attention [88,89]. These types of classifiers could possibly overcome some of the limitations of the current algorithm. However, they also involve a higher computational load, making them less suitable for embedded software. Additionally, their interpretation is less intuitive than the current adjustable algorithm. Future research should explore the current state of algorithm development in order to achieve optimal PA classification in hospitalized patients. Another possibility to improve the classification of PA in hospitalized patients may be the use of multiple accelerometers. However, this is not practical in a clinical setting, requires more resources, and may adversely affect compliance [42,60,90].

As we included a range of different performance metrics, we have not only provided a complete overview of the performance of the algorithm, but also enabled comparisons with others studies. Nevertheless, comparing the results of the current study with those of other validation studies is challenging, due to differences in the validation protocols, patient populations, accelerometer types, wear locations, and performance metrics used. Additionally, most studies have not transparently reported their classification algorithms, as these are often proprietary and not disclosed [35,46,48]. Out of seven studies, only Lipperts et al. and Pedersen et al. have transparently described their classification algorithms [12,35,46,48,55,61,62].

Most previous studies investigating the validity of accelerometers in hospitalized patients were able to correctly classify sedentary (lying and/or sitting) activities [12,46,48,55,61], and all studies were able to correctly detect postural transitions [46,55,61]. However, they all experienced difficulties in accurately classifying standing and/or walking activities, independent of their wear locations or study protocols [12,35,46,48,55]. Brown et al. and Pedersen et al. were both unable to differentiate standing from walking in their respective samples of 39 and 6 acute medical patients aged 65 years or older. Brown et al. validated their algorithm using a free-living protocol with an accelerometer worn at the ankle, while Pedersen et al. used a standardized protocol with two accelerometers, one worn at the ankle and one on the upper leg. Neither used post-hoc video analysis as a reference, nor did they investigate the validity of the algorithm to detect postural transitions [12,48]. Valkenet et al. investigated the validity of three accelerometers, each with a different algorithm and wear location (i.e., hip, upper thigh, and lumbar waist). Although the classification of walking showed good sensitivity values (90 to 95%) for all three wear locations, the classification of standing, sitting and lying showed lower sensitivity values, ranging between 13 and 79%, 57 and 94%, and 0 and 79%, respectively. However, the validation was performed with only two inpatients using a standardized protocol, and the validity of the algorithms to detect postural transitions was not investigated [35]. Baldwin et al. investigated the validity of an accelerometer worn at the thigh in eight patients recovering from a critical illness. Although the validation was performed using a free-living protocol and the validity of the algorithm to detect postural transitions was investigated, direct observation by only one observer was used as a reference. The results showed an overestimation of the time spent standing and an underestimation of the time spent walking. With median (interquartile range) APE values of 21.9% (101.1%) for time spent standing and 18.7% (73.1%) for time spent walking, both values exceeded our acceptable limit of 10% [46]. Although the median (Q1 to Q3) APE of 29.2% (14.6% to 55.2%) for standing activity found for HFITAL also exceeds this limit, walking was detected more accurately by HFITAL. Lastly, Lipperts et al. investigated the validity of an accelerometer worn at the lateral side of the unaffected leg, using a validation protocol approaching free-living conditions in 40 patients who underwent total joint arthroplasty 3–14 days prior to participation. Their results showed accuracy values above 92% for the classification of sitting, standing, level walking, stair climbing, and cycling activities and a mean error of duration of 2.9% for standing. As in our study, they found an underestimation of average standing duration and an overestimation of average walking and sitting duration [55]. However, as the patients included in our study had undergone total joint arthroplasty 1–2 days prior to participation, they can be assumed to have walked at a lower walking speed and with a more impaired gait pattern, which made it more challenging for HFITAL to correctly classify standing and dynamic activities. Taking into account that the current study was performed under free-living conditions in a population in which impaired and slow gait were common, the performance metrics of HFITAL are at least similar to, or possibly even better than, those reported by other validation studies.

A strength of our study is that the optimization and validation of the algorithm were performed in acutely hospitalized elderly patients and orthopedic patients following elective TKA or THA. These groups were deliberately chosen as they tend to walk very slowly, often with an impaired gait or walking aid, and therefore make accurate classification of standing and dynamic activities more challenging. The accelerometer is intended to be used in a wider variety of hospitalized patients, and we expect the performance metrics to be better when used in other patient populations. Second, our optimized algorithm and validation methodology were transparently described, enabling researchers and clinicians to compare the algorithm and results with other studies [49,51,91]. Third, video recordings were used as a gold standard, with a good inter-observer reliability for the classification of all activities (ICC ≥ 0.9). Fourth, the performance metrics of HFITAL were comparable for the optimization and validation groups, indicating a consistent performance of HFITAL when used to classify the PA behavior of patients outside the optimization group. Lastly, the validation of the optimized algorithm was performed under free-living conditions, providing a more accurate indication of the actual performance of the algorithm [55,61]. As physical therapy sessions often comprise a significant part of patients’ PA behavior during hospitalization, we chose to perform the validation during these sessions. This also ensured that sufficient time was spent performing standing and dynamic activities without consuming too much of the patients’ time, thereby avoiding practical and ethical difficulties.

There are also some limitations to the current study that should be addressed. First, the physical therapists may have given patients instructions regarding their gait pattern or walking speed, thereby influencing natural conditions. Second, the duration of the validation protocol was influenced by the duration of the physical therapy session, resulting in shorter measurement periods than anticipated. However, a compromise had to be made between capturing sufficient time spent performing standing and dynamic activities and the duration of the free-living validation protocol. Third, walking speed was not assessed because the validation was performed under free-living conditions. This could, however, have enabled us to investigate a possible relationship between walking speed and the ability of the algorithm to classify standing and dynamic activities within acceptable limits.

Our study has some important implications for clinical practice. As hospitalized patients need to increase their amount of PA and break up prolonged periods of sedentary activity, the classification of dynamic activity and the detection of postural transitions are considered the most important outcome measures for PA monitoring [1,35,45]. The results show that although HFITAL is not able to classify standing activity accurately, it is able to validly classify sedentary and dynamic activities as well as postural transitions in hospitalized patients under free-living conditions. With performance metrics that are similar, or even better, than those of existing algorithms, HFITAL proves to be a good alternative. Moreover, HFITAL can be embedded in eHealth applications, such as Hospital Fit [1]. As the algorithm involves a relatively low computational load, it is suitable to be embedded in an accelerometer without reducing its battery life. Embedding HFITAL in Hospital Fit will improve continuous PA monitoring with real-time feedback as a part of standard care. This will provide patients and healthcare professionals with more accurate feedback, enabling optimal support for patients’ PA behavior and recovery.

## 5. Conclusions

The optimized PA classification algorithm (HFITAL) is able to validly classify sedentary and dynamic activities as well as to detect postural transitions under free-living conditions in hospitalized patients with an accelerometer worn on the upper leg. As hospitalized patients need to increase their amount of PA and interrupt prolonged periods of sedentary activity, HFITAL is a suitable algorithm to classify PA in these patients. In order to improve PA monitoring as a part of standard care and improve recovery in hospitalized patients, we propose to embed HFITAL in eHealth applications, such as Hospital Fit.

## Figures and Tables

**Figure 1 sensors-21-01652-f001:**
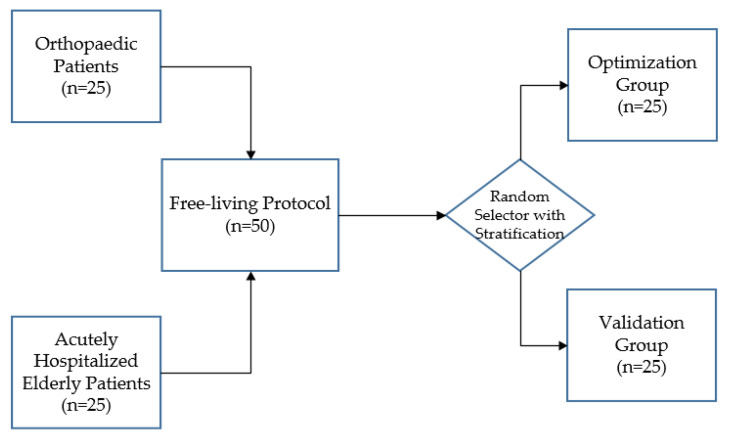
After stratification according to department, 50 patients were randomly assigned to the optimization or validation group.

**Figure 2 sensors-21-01652-f002:**
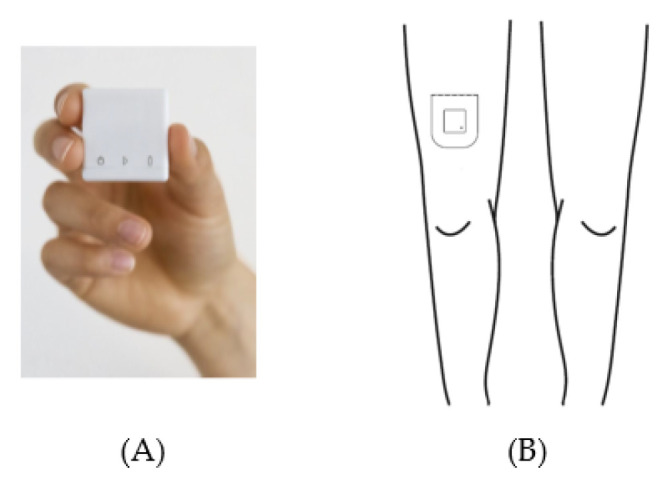
The MOX activity logger (**A**) and the wear location on the upper leg (**B**).

**Figure 3 sensors-21-01652-f003:**
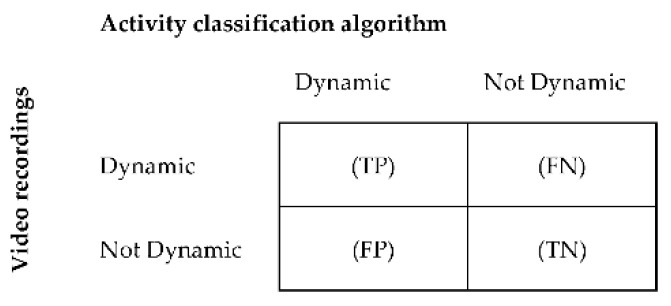
Binary confusion matrix for the classification of dynamic activities per patient. True positive (TP) = number of windows correctly classified by the algorithm as dynamic; false positive (FP) = number of windows incorrectly classified as dynamic; true negative (TN) = number of windows correctly classified as not dynamic, and false negative (FN) = number of windows incorrectly classified as not dynamic.

**Figure 4 sensors-21-01652-f004:**
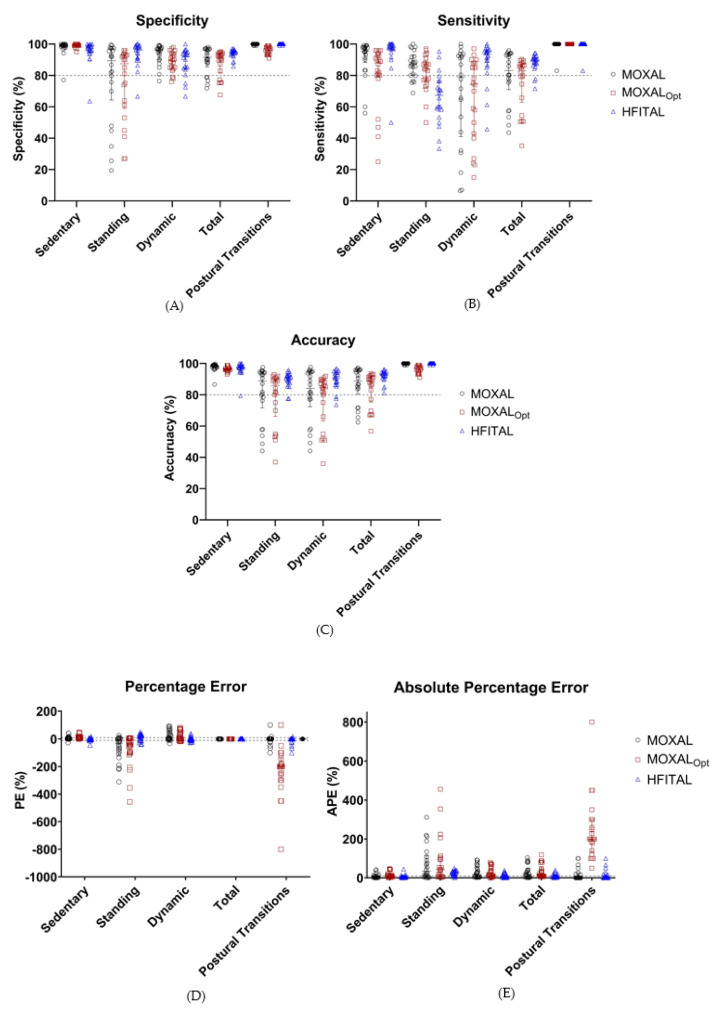
Performance metrics (%, Sensitivity (**A**), specificity (**B**), accuracy (**C**), percentage error (**D**), and absolute percentage error (**E**)) of the classification of activities by MOXAL, MOXAL_Opt_, and HFITAL within the optimization group. All individual values are shown. Acceptable limits are represented by dashed lines. MOXAL is represented in black, MOXAL_Opt_ in brown, and HFITAL in blue. (MOXAL = adjustable classification algorithm validated in community-dwelling healthy elderly persons with an upper leg wear location, used as the starting point for the optimization process. MOXAL_Opt_ = classification algorithm after optimization of three adjustable parameter settings of MOXAL to reduce the absolute percentage error for total activity. HFITAL = classification algorithm after additional modifications were introduced to MOXAL regarding the decision tree and the calculation of the amount of physical activity).

**Figure 5 sensors-21-01652-f005:**
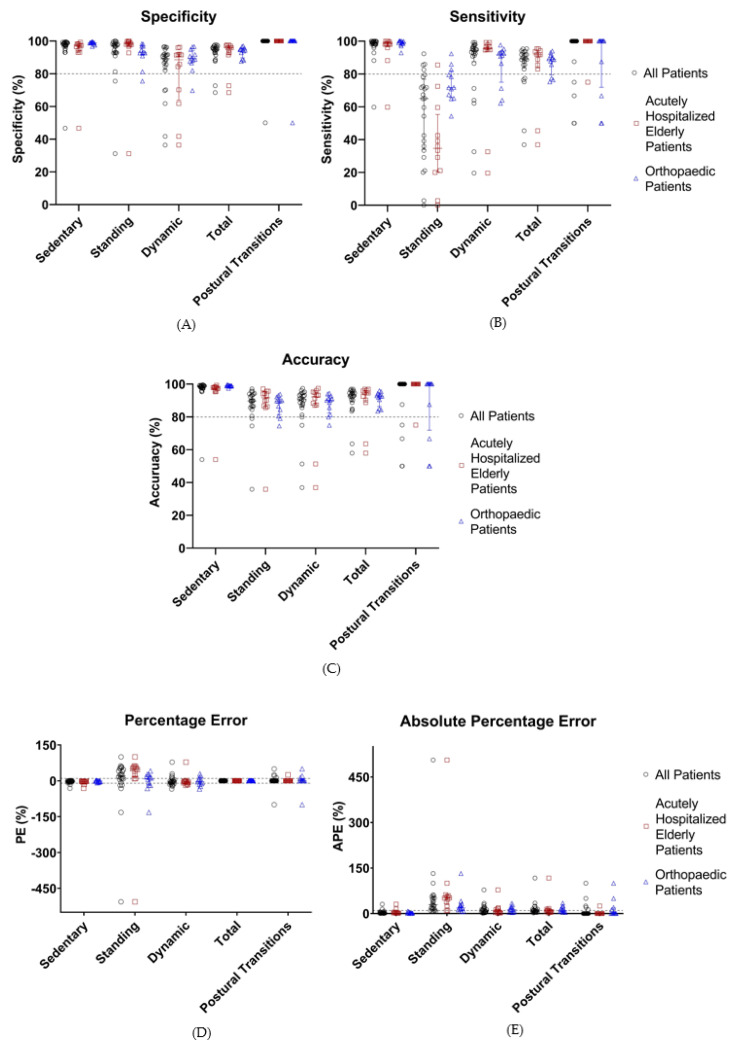
Performance metrics (%, sensitivity (**A**), specificity (**B**), accuracy (**C**), percentage error (**D**), and absolute percentage error (**E**)) of the classification of activities by HFITAL within the validation group. All individual values are shown. Acceptable limits are represented by dashed lines. “All Patients” are represented in black, “Acutely Hospitalized Elderly Patients” in brown, and “Orthopedic Patients” in blue. (MOXAL = adjustable classification algorithm validated in community-dwelling healthy elderly persons with an upper leg wear location, used as the starting point for the optimization process. MOXAL_Opt_ = classification algorithm after optimization of three adjustable parameter settings of MOXAL to reduce the absolute percentage error for total activity. HFITAL = classification algorithm after additional modifications were introduced to MOXAL regarding the decision tree and the calculation of the amount of physical activity).

**Table 1 sensors-21-01652-t001:** Definitions for activity classification of the video recordings.

Activity	Definition
Sedentary	Patient is in a seated or lying position (angle between upper leg and gravity vector < 60 degrees)
Standing	Patient is in an upright position (angle between upper leg and gravity vector > 60 degrees) for more than 2 s without activity of the lower extremities
Dynamic	Patient performs physical activity with the lower extremities for at least 2 s, such as walking, stair climbing, or cycling
Postural Transitions	Transition from a sedentary activity to a standing or dynamic activity

**Table 2 sensors-21-01652-t002:** Characteristics of study participants in the optimization and validation groups.

	Optimization Group			Validation Group		
Characteristic	All Patients (*n* = 22)	Acutely Hospitalized Elderly Patients (*n* = 11)	Orthopedic Patients (*n* = 11)	All Patients (*n* = 24)	Acutely Hospitalized Elderly Patients (*n* = 12)	Orthopedic Patients (*n* = 12)
Sex, female (*n*, %)	7 (31.8%)	2 (18.2%)	5 (45.5%)	14 (58.3%)	7 (58.3%)	7 (58.3%)
Age, years (median, Q1 to Q3)	75.4 (72.6 to 82.0)	82.0 (75.4 to 87.7)	73.7 (66.4 to 76.0)	75.8 (70.3 to 85.5)	84.7 (77.0 to 88.3)	70.1 (61.2 to 75.5)
Walking Aid (*n*, %)	20 (90.9%)	11 (100.0%)	9 (81.8%)	21 (87.5%)	12 (100.0%)	9 (75.0%)

**Table 3 sensors-21-01652-t003:** Median values (Q1 to Q3) of the performance metrics (%, sensitivity, specificity, accuracy, PE, and APE) of the classification of activities by MOXAL, MOXAL_Opt_, and HFITAL within the optimization group (*n* = 22).

Activity	Algorithm	Sensitivity (%)	Specificity (%)	Accuracy (%)	PE (%)	APE (%)
Sedentary	MOXAL	95.6 (88.3 to 98.0)	99.1 (98.1 to 99.6)	98.5 (97.4 to 98.9)	2.6 (0.5 to 8.7)	3.2 (1.0 to 11.4)
	MOXAL_Opt_	88.0 (79.5 to 93.3)	99.0 (99.0 to 100.0)	96.0 (96.0 to 97.0)	10.4 (6.1 to 17.4)	10.4 (6.1 to 17.4)
	HFITAL	97.7 (96.2 to 100.0)	97.1 (95.2 to 99.2)	97.5 (95.3 to 98.9)	−2.1 (−4.4 to 3.6)	3.8 (2.3 to 7.2)
Standing	MOXAL	87.4 (79.9 to 93.1)	89.4 (64.4 to 97.3)	88.8 (71.6 to 94.4)	−33.1(−114.8 to 1.1)	34.0 (6.1 to 114.8)
	MOXAL_Opt_	84.0 (76.3 to 88.8)	87.5 (59.0 to 93.0)	86.0 (66.3 to 90.3)	−42.9 (−106.8 to 1.4)	42.9 (8.0 to 106.8)
	HFITAL	67.3 (57.1 to 76.4)	96.8 (90.5 to 98.9)	90.3 (87.3 to 92.5)	20.2 (−10.1 to 30.5)	25.1 (11.8 to 35.5)
Dynamic	MOXAL	79.0 (40.1 to 92.9)	96.1 (90.2 to 97.5)	84.1 (72.3 to 94.6)	8.8 (−2.6 to 55.4)	18.2 (3.4 to 55.4)
	MOXAL_Opt_	74.5 (42.3 to 88.0)	89.5 (84.8 to 94.0)	84.5 (63.4 to 89.0)	8.7 (−9.0 to 50.7)	17.5 (8.7 to 56.0)
	HFITAL	93.6 (85.9 to 96.2)	92.2 (84.4 to 94.9)	90.9 (86.5 to 94.2)	−3.2 (−8.2 to 4.7)	6.9 (3.1 to 16.8)
Total	MOXAL	83.2 (71.0 to 93.8)	91.6 (85.5 to 96.9)	88.8 (80.7 to 95.8)	0.1 (−0.1 to 0.3)	18.9 (4.2 to 51.0)
	MOXAL_Opt_	82.9 (62.9 to 87.3)	91.4 (81.5 to 93.6)	88.6 (75.3 to 91.5)	−0.1 (−0.1 to 0.1)	11.8 (8.7 to 56.0)
	HFITAL	89.7 (86.1 to 91.5)	94.8 (93.1 to 95.7)	93.1 (90.7 to 94.3)	0.2 (−0.1 to 0.4)	7.6 (4.8 to 15.3)
Postural Transitions	MOXAL	100.0 (100.0 to 100.0)	100.0 (100.0 to 100.0)	100.0 (100.0 to 100)	0.0 (0.0 to 0.0)	0.0 (0.0 to 19.0)
	MOXAL_Opt_	100.0 (100.0 to 100.0)	96.1 (93.6 to 98.1)	96.2 (93.8 to 98.1)	−200.0 (−290.0 to −150.0)	200.0 (150.0 to 190.0)
	HFITAL	100.0 (100.0 to 100.0)	100.0 (100.0 to 100.0)	100.0 (100.0 to 100.0)	0.0 (0.0 to 0.0)	0.0 (0.0 to 13.0)

PE = percentage error. APE = absolute percentage error. MOXAL = adjustable classification algorithm validated in community-dwelling healthy elderly persons with an upper leg wear location, used as the starting point for the optimization process. MOXAL_Opt_ = classification algorithm after optimization of three adjustable parameter settings of MOXAL to reduce absolute percentage error for total activity. HFITAL = classification algorithm after additional modifications were introduced to MOXAL regarding the decision tree and the calculation of the amount of physical activity.

**Table 4 sensors-21-01652-t004:** Median values (Q1 to Q3) of the performance metrics (%, sensitivity, specificity, accuracy, PE, and APE) of the classification of activities by HFITAL within all patients of the validation group (*n* = 24) and the subgroups of acutely hospitalized elderly patients (*n* = 12) and orthopedic patients (*n* = 12).

Activity	Population	Sensitivity (%)	Specificity (%)	Accuracy (%)	PE (%)	APE (%)
Sedentary	All Patients	98.7 (98.0 to 100.0)	98.2 (96.6 to 98.9)	98.5 (97.4 to 99.1)	−1.9 (−4.9 to −0.7)	1.9 (0.7 to 4.8)
	Acutely Hospitalized Elderly Patients	98.3 (96.5 to 99.9)	96.9 (93.4 to 98.2)	97.6 (95.6 to 98.1)	−2.1 (−5.4 to −1.7)	2.1 (1.7 to 5.4)
	Orthopedic Patients	99.3 (98.2 to 100.0)	98.9 (98.3 to 99.3)	98.9 (98.8 to 99.3)	−0.8 (−3.6 to 0.2)	0.8 (0.5 to 3.6)
Standing	All Patients	65.0 (34.1 to 76.9)	96.9 (92.7 to 98.5)	89.8 (85.8 to 93.7)	21.3 (−3.9 to 50.2)	29.2 (14.6 to 55.2)
	Acutely Hospitalized Elderly Patients	34.7 (20.3 to 55.3)	98.3 (96.9 to 99.7)	91.7 (86.4 to 95.4)	49.0 (13.6 to 58.6)	51.6 (29.2 to 61.2)
	Orthopedic Patients	71.8 (65.7 to 81.8)	93.1 (91.4 to 96.9)	89.6 (81.5 to 91.6)	9.1 (−18.5 to 23.7)	18.2 (12.8 to 30.4)
Dynamic	All Patients	94.3 (87.5 to 96.5)	89.2 (82.6 to 91.9)	90.5 (85.9 to 93.8)	−4.2 (−12.5 to 3.1)	8.6 (4.0 to 18.2)
	Acutely Hospitalized Elderly Patients	95.6 (94.6 to 97.9)	88.6 (63.9 to 91.9)	92.2 (87.1 to 95.1)	−5.1 (−11.6 to −1.6)	6.9 (2.2 to 15.4)
	Orthopedic Patients	91.9 (75.0 to 93.7)	89.2 (86.8 to 94.1)	90.1 (82.3 to 92.3)	−3.6 (−13.6 to 13.7)	12.9 (4.9 to 22.1)
Total	All Patients	89.2 (83.6 to 92.8)	94.6 (91.8 to 96.4)	92.8 (89.1 to 95.2)	0.2 (0.0 to 0.4)	8.6 (5.3 to 14.7)
	Acutely Hospitalized Elderly Patients	91.8 (83.6 to 94.1)	95.9 (91.8 to 97.1)	94.5 (89.1 to 96.1)	0.2 (0.0 to 0.3)	8.2 (4.5 to 13.7)
	Orthopedic Patients	88.9 (79.5 to 91.1)	94.5 (89.8 to 95.5)	92.6 (86.3 to 94.1)	0.3 (0.1 to 0.5)	8.6 (6.6 to 21.4)
Postural Transitions	All Patients	100.0 (100.0 to 100.0)	100.0 (100.0 to 100.0)	100.0 (100.0 to 100.0)	0.0 (0.0 to 0.0)	0.0 (0.0 to 0.0)
	Acutely Hospitalized Elderly Patients	100.0 (100.0 to 100.0)	100.0 (100.0 to 100.0)	100.0 (100.0 to 100.0)	0.0 (0.0 to 0.0)	0.0 (0.0 to 0.0)
	Orthopedic Patients	100.0 (82.3 to 100.0)	100.0 (100.0 to 100.0)	100.0 (82.3 to 100.0)	0.0 (0.0 to 3.1)	0.0 (0.0 to 14.4)

PE = percentage error. APE = absolute percentage error. MOXAL = adjustable classification algorithm validated in community-dwelling healthy elderly persons with an upper leg wear location, used as the starting point for the optimization process. MOXAL_Opt_ = classification algorithm after optimization of three adjustable parameter settings of MOXAL to reduce the absolute percentage error for total activity. HFITAL = classification algorithm after additional modifications were introduced to MOXAL regarding the decision tree and the calculation of the amount of physical activity.

## Data Availability

Essential data relevant to the findings of this study can be found in the Appendix A. Additional data supporting the findings of this work are available upon reasonable request by contacting the corresponding author.

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
