# Peer review of "Optimization and Validation of a Classification Algorithm for Assessment of Physical Activity in Hospitalized Patients"

_sensors, 2021, doi:10.3390/s21051652_

Round 1

Reviewer 1 Report

The study aim was to optimize and validate a PA classification algorithm that discriminates between sedentary, standing, and dynamic activities in hospitalized patients under free living conditions. Overall, this was well thought out study with a strong intro/review of literature and identified a strong need for the study. I feel the manuscript met the aims in a clear and succinct way and discussed the different issues directly. There were some questions and unclear areas within the different sections, but overall, I feel that the manuscript is adding to the value of measuring sedentary to active hospitalized patients under free living conditions. I have outlined by line number the different questions and unclear sections that need to be addressed below. Line 119 How was written informed consent obtained? Line 120-Explain how data was stored securely and anonymity was guaranteed. Line 135-How did you come up with this sample size? G power estimations? I understand that you captured between 8 to 99 participants based on other studies, but G power estimation is a sound way of knowing you have enough participants to create a valid population estimate for the analyses you are doing. Line 138 - How was stratification by the department determined? Line 140-How did you randomize the participants? What tool did you use? Lines 148-150- You state that you used a range between the first and last physical therapy sessions with your patients. Would this impact the results if you have some participants near the end of their PT programs and others just starting their programs? Meaning wouldn’t they display more regular physical activity and walking patterns towards the end of their PT sessions versus those who are just starting PT? Lines 170-172-Please clarify why the accelerometer is placed on the leg Lines 191-294-At times you state you use different data processing instruments and use acronyms such as HFITAL or BORIS. Please explain exactly what those programs are to make it clearer for the reader. Line 302 - Table 1. Try to make all the lines of the table uniform and not spread in two rows (i.e. 14 (58.3%) Lines 309-320-At times you use a double parenthesis. I don't think they should be there. Lines 383-Please clarify why you are using the median and not mean scores References-Several references are quite old. They seem out of date to me. Please either find more recent references or don't use them unless they are landmark studies.

Reviewer 2 Report

The authors investigate the validation and optimization classification algorithm for assessing physical activity in Hospitalized patients.

The proposed study is interesting but there are some points that the authors should better discuss.

The authors should be better described the novelties of their study with respect to existing ones. In particular, the author should discuss limitation and cons of the examined approaches. Furthermore, the authors should provide more details and discussion about the obtained results. The Discussion section also needs to be improved by analyzing the outcome of evaluation section.

I suggest to further analyze more recent approaches about the examined topics. In particular, I suggest the following papers to further investigate graph-based machine learning and influence diffusion approaches for COVID pandemic in the introduction section:

1) An Epidemiological Neural network exploiting Dynamic Graph Structured Data applied to the COVID-19 outbreak. IEEE Transactions on Big Data.

2) Multimedia social network modeling: A proposal. In 2016 IEEE Tenth International Conference on Semantic Computing, 2016; pp.448-453.

Finally, I suggest to perform a linguistic revision.

Reviewer 3 Report

BRIEF SUMMARY

The study aimed to was to optimize and validate (in comparison to video analysis) a classification algorithm (based on acceleration data) that discriminates between sedentary, standing and dynamic activities in hospitalized patients under real-life conditions. Authors present optimal parameter setting for the algorithm and demonstrate acceptable validity for most activities, except for the classification of standing activity.

I congratulate authors on their work. This is a well-written and technically sound paper with informative figures and tables. Overall, I found the topic timely and clinically important. Minor suggestions are listed below.

SPECIFIC COMMENTS

ABSTRACT

  1. Lines 31-32: “the algorithm can assist in improving patients’ PA behavior.” I find this too far-fetched. I would suggest to be more specific and present only direct applications.

INTRODUCTION

  1. Please add in the introduction few sentences regarding advantages wearable sensors/accelerometers provide for measuring physical activity over self-reported measures. Crucial point that is currently lacking here.

METHODS

  1. Please state whether the MOX Activity Logger has an established validity and reliability.
  2. Line 192: Please make a reference where readers can find the definitions of these activities
  3. Table A1” Please introduce first what A means. This becomes only evident at the end of the paper (as in point 4).
  4. Can authors provide a figure (here or in the appendices) with raw acceleration data from one example subject, depicting each activity classification?
  5. Subsection 2.4.3. Can author provide more details regarding how exactly did they validate the algorithm against the video analysis? This might not be clear for a non-specialist audience.

RESULTS

  1. Please improve the graphics of the Table 1
  2. Tables: Please provide (in the tables legends at the bottom) what MOXAL, MOXALOpt and HFITAL mean.

CONCLUSIONS

  1. Conclusions are slightly different form the one presented in the abstract. Please modify accounting for my previous comment in point 1.

Round 2

Reviewer 2 Report

I think that the authors have addressed all my concerns

Reviewer 3 Report

Thank you, authors responded to my comments appropriately.